# Molecular Tracking of the Origin of Vesicular Stomatitis Outbreaks in 2004 and 2018, Ecuador

**DOI:** 10.3390/vetsci10030181

**Published:** 2023-02-24

**Authors:** David Vasco-Julio, Dayana Aguilar, Alexander Maldonado, Euclides de la Torre, Maria Soledad Cisneros-Montufar, Carlos Bastidas-Caldes, Juan-Carlos Navarro, Jacobus H. de Waard

**Affiliations:** 1One Health Research Group, Facultad de Ingenierías y Ciencias Aplicadas, Carrera de Ingeniería en Biotecnología, Universidad de Las Américas, Quito 170530, Ecuador; 2Programa de Posgrado en Ciencias Biológicas, Universidad Nacional Autónoma de México, Ciudad de México 04510, Mexico; 3Centro de Investigación Sobre Enfermedades Infecciosas Instituto Nacional de Salud Pública, Cuernavaca 62050, Mexico; 4One Health Research Group, Facultad de Ciencias Veterinarias, Universidad de Las Américas, Quito 170530, Ecuador; 5Escuela de Medicina, Facultad de Ciencias Médicas, de la Salud y de la Vida, Universidad Internacional del Ecuador, Quito 170113, Ecuador; 6Dirección de Diagnóstico Animal, Agencia de Regulación y Control Fito y Zoosanitario, Agrocalidad, Quito 170518, Ecuador; 7Group of Emerging and Neglected Diseases, Ecoepidemiology and Biodiversity, Health Sciences Faculty, Universidad Internacional SEK, Quito 170521, Ecuador; 8Programa de Doctorado en Salud Pública y Animal, Universidad de Extremadura, 06006 Badajoz, Spain

**Keywords:** vesicular stomatitis virus, molecular epidemiology, outbreak, phylogeny

## Abstract

**Simple Summary:**

Vesicular stomatitis (VS) is a viral disease that primarily affects cattle, horses and swine. In affected livestock, the VS virus (VSV) causes painful blister-like lesions in the mouth, on the tongue, lips, nostrils, hooves, and teats and infected animals usually refuse to eat and drink, causing livestock production losses. VS is also a zoonotic disease and can cause influenza-like symptoms in humans. In 2018, Ecuador experienced an outbreak of VS affecting cattle in most of the provinces of the country. Here we determined the phylogenetic relationship among the virus strains isolated in that year. Our analysis suggests different transmission patterns in the major geographic regions of Ecuador; several small and independent outbreaks, most probably transmitted by vectors in the Amazon, and an outbreak caused by the movement of livestock and/or fomites in the Andean and Coastal regions. For better control of future outbreaks, we recommend more research into vectors and vertebrate reservoirs in Ecuador. This can further elucidate the mechanisms of the emergence and re-emergence of the virus.

**Abstract:**

The Vesicular Stomatitis Virus (VSV) is an arbovirus causing vesicular stomatitis (VS) in livestock. There are two serotypes recognized: New Jersey (VSNJV) and Indiana (VSIV). The virus can be transmitted directly by contact or by vectors. In 2018, Ecuador experienced an outbreak of Vesicular Stomatitis (VS) in cattle, caused by VSNJV and VSVIV, with 399 cases reported distributed over 18 provinces. We determined the phylogenetic relationships among 67 strains. For the construction of phylogenetic trees, the viral phosphoprotein gene was sequenced, and trees were constructed based on the Maximum Likelihood method using 2004 outbreak strains from Ecuador (GenBank) and the 2018 sequences (this article). We built a haplotype network for VSNJV to trace the origin of the 2004 and 2018 epizootics through topology and mutation connections. These analyses suggest two different origins, one related to the 2004 outbreak and the other from a transmission source in 2018. Our analysis also suggests different transmission patterns; several small and independent outbreaks, most probably transmitted by vectors in the Amazon, and another outbreak caused by the movement of livestock in the Andean and Coastal regions. We recommend further research into vectors and vertebrate reservoirs in Ecuador to clarify the mechanisms of the reemergence of the virus.

## 1. Introduction

Vesicular stomatitis virus (VSV) is an RNA arbovirus [1] belonging to the Rhabdoviridae family and the *Vesiculovirus* genus [2] and causes an acute, contagious vesicular disease. To date, two VSV serotypes have been identified: Indiana (IN) and New Jersey (NJ) [3]. Of these two serotypes, VSIV has three subtypes: VSIV 1 (Indiana 1), VSIV 3 or VSV-AV (Indiana 3 or Alagoas), and VSIV 2 (Indiana 2 or Cocal) [4]. The serotypes that affect North, Central, and South America the most are VSNJV and VSIV 1 [5], although VSIV 3 prevails in Brazil, and VSIV 2 has been reported in Brazil and Argentina [6]. 

The usual hosts of VSV are generally mammals, such as cattle, horses, and pigs, but also arthropods and occasionally humans. This viral infection can be considered a minor zoonosis [7]. Although vesicular stomatitis can be transmitted by direct contact and fomites, insects are believed to play a major role in transmission between animals [8]. The insects that carry VSV are known to include sandflies, black flies, and biting midges. The ability of VSV to be transmitted through vectors is a great problem, since these vectors do not function only as simple means of transport, but also as a virus reservoir [6,9]. Vector control, therefore, should form an integral part of controlling vesicular stomatitis outbreaks. In Ecuador, vesicular stomatitis is a disease whose control is mainly based on quarantines and animal movement restrictions. Occasionally, a vaccine is used to try to reduce viral spread, namely a bivalent vaccine manufactured by VECOL, Colombia, which stimulates the production of specific antibodies [10].

Vesicular stomatitis virus causes huge economic losses in cattle because of a considerable decrease in milk and meat production [11]. In addition, because its clinical manifestations are extremely similar to those of foot-and-mouth disease, outbreaks of VS result in the closing of borders, livestock travel restrictions, and quarantine measures [12]. Concerning the clinical signs of VS, after an initial febrile period, characteristic vesicular lesions occur on the muzzle, lips, tongue, ears, sheath, udder, ventral abdomen, and/or coronary bands of the hoof [13]. The incubation period varies from 2–6 days with an average of 3–5 days in cattle, and visible vesicular lesions can appear 24 h after infection. The first sign of illness is often excessive salivation. In humans, the incubation period varies from 24 h to 6 days [14]. 

Between January and December 2018, an outbreak of VS occurred in Ecuador with a total of 399 confirmed cases throughout 18 provinces. This outbreak has been described by Salinas et al. in an earlier publication [15]. The main objective of their study was to characterize the temporal and spatial dynamics of the prevalence of VS. Their study determined that the highest number of VS cases happened between weeks 35 and 44. Concerning the etiological agent, 93.23% of the cases were positive for VSV-NJ, 5.52% for VSIV, and 1.25% for both serotypes. Similarly, it was noted that the outbreaks emerged at the end of the dry season and the beginning of the rainy season. It is, therefore, believed that this outbreak was caused by the movement of vectors. However, the origin of the outbreak remains unknown. The rapid spread of the outbreak within a few weeks throughout the whole country and over several hundreds of kilometers indicates probably more than one vector and different transmission mechanisms. Our study aimed to trace the origin of the outbreak and to determine transmission patterns through the construction of phylogenetic trees of the 2018 isolates and strains from an outbreak in Ecuador in 2004 [16]. For this purpose, we used the partial nucleotide sequences of the hypervariable region of the phosphoprotein P of the virus that has been used extensively for VSV-NJ and VSIV phylogenetic analysis.

## 2. Materials and Methods

### 2.1. Virus Isolation in Cell Culture 

To obtain enough viral material, a cell culture of the isolates previously stored in Gibco™ DMEM (Thermo Fisher Scientific, Waltham, MA, USA) culture medium and stored in liquid nitrogen was initiated. The viruses were grown following the protocol established by the OIE Manual of Diagnostic Tests and Vaccines for Terrestrial Animals 2021, Chapter 3.1.23: Vesicular stomatitis (Version adopted in May 2021) [17]. Briefly, the swabs with the VSV strains, stored in liquid nitrogen, were inoculated into cell culture vessels with BHK-21 cells and incubated at 37 °C for 1 h [absorption]. After washing the cell cultures three times with cell culture medium, the cell culture medium was replaced with medium containing 2.5% fetal bovine serum (FBS) and incubated at 35–37 °C and examined for cytopathic effect (CPE) after 24, 48 and 72 h. If visible CPE was detected a clarified medium of infected cells was frozen and used for RNA isolation. If, after 72 h, no CPE had been detected, the virus was considered dead. 

### 2.2. RNA Extraction and Reverse Transcriptase-PCR (RT-PCR)

For viral RNA extraction, the AccuPrep^®^ Viral RNA Extraction Kit (Bioneer, South Korea) was used following the protocol established by the manufacturer. To carry out the RT-PCR, two pairs of primers that amplified the VSV phosphoprotein gene (protein P) were used. The primers for VSNJV spanned a region of the gene between position 102 (P-NJ 102; 5′ GAGAGGATAAATATCTCC 3′) and 744 (P-NJ 744; 5′ GGGCATACTGAAGAATA 3′) and resulted in a fragment of 642 base pairs. The VSIV primers amplified a region between position 179 (P-Ind 179; 5′ GCAGATGATTCTGACAC 3′) and 793 (P-Ind 793; 5′ GACTCT(C/T)GCCTG(A/ G) TTGTA 3′) and resulted in a fragment of 614 bp [16].

For the RT-PCR reaction, the MegaFi™ One-Step RT-PCR kit (ABM) was used, consisting of the following concentrations for a final volume of 15 μL: 0.3 μM of each primer, 1 ng/μL of RNA, 1X of One-Step RT-PCR Buffer, and 1U/μL RT-PCR Enzyme Mix. For reverse transcription, cDNA synthesis was performed at 60 °C for 15 min. The amplification procedure started with an initial denaturation at 98 °C for 30 s, followed by 35 cycles at 98 °C for 10 s, 52 °C for 30 s, 72 °C for 30 s, and a final extension of 2 min at 72 °C.

RT-PCR products were analyzed by electrophoresis using a 1.5% agarose gel with SYBR Safe for visualization and 1X TBE as the buffer. Electrophoresis was run at 100 V for 30 min in a Labnet Enduro Gel XL horizontal chamber (Labnet International, Inc., New York City, NY, USA). The gel was visualized on a Chem-iDocTM Imaging Systems (BioRad, Hercules, CA, USA) photographic documentation system and using Image LabTM software (BioRad, USA). The positive fragments were sequenced with Sanger sequencing with an ABI 3500xL Genetic Analyzer (Applied Biosystems, Waltham, MA, USA) in a BigDye 3.1^®^ capillary electrophoresis matrix. 

### 2.3. Phylogenetic Analysis

#### 2.3.1. Construction of Phylogenetic Trees

For the generation of a consensus sequence of the partial P protein sequence of the VSNJV and VSIV isolates, forward and reverse sequences were aligned using the UniPro Gene program. Sequence contig reconstruction was performed using Assembler by MacVector software 17.5.5. [18]. The sequence identity was confirmed by BLAST in NCBI resources. A total of 68 sequences of VSV from the United States, Mexico, Honduras, Panama, Costa Rica, Nicaragua, Guatemala, Argentina, Brazil, El Salvador, Colombia, and Ecuador were retrieved from GenBank (All accession numbers, sample origin, serotypes, and isolates are described in Appendix A) [16,19,20,21,22,23,24,25,26,27,28,29]. For the VSNJV analysis, we included as an outgroup two sequences of VSIV: an Indiana strain from Ecuador and one from Costa Rica with accession numbers EF028138.1 and EF028137.1, respectively. The initial alignment matrix consisted of 614 bp and was trimmed to obtain a final alignment of 410 bp VSIV matrix. The VSVNJ sequences in the alignment matrix were trimmed to 422 pb from an initial alignment of 642 pb. The New Jersey and Indiana strains were separately analyzed. For VSIV analysis two of our VSNJV sequences (ON567174 and ON567120) were included as an outgroup. DNA sequences were aligned using MacVector 17.5.5 by the ClustalW algorithm with high gap creation and extension penalties by 30.0 and 10.0, respectively, searching for a strong positional homology. 

Finally, the phylogenetic trees of VSNJV and VSIV isolates from Ecuador were built from the 2004 and 2018 isolates by using the maximum parsimony (MP) (100 replicates, reweighted under Rescaled Consistency index, RC index) using PAUP 4.0a [30] and maximum likelihood (ML) analyses conducted in MEGA X [31]. The ML tree was corrected with the Tamura 3-parameter model [32]. The final trees for VSNJV and VSIV were edited using FigTree 1.4.4 [33]. The robustness values for all the analyses were estimated using bootstrapping with 500 pseudo-replicates and are shown as percentages. A bootstrap re-sample with 1000 pseudo-replicates did not show significant differences in the values of the resulting topologies. The strain number of the 2018 samples, the GenBank accession number, the geographic origin in Ecuador, the month of isolation, and the serotype can be found in Appendix A together with the accession numbers of the sequence of the P gene from the outbreak in 2004 in Ecuador. 

#### 2.3.2. Haplotype Network of the VSNJV Sequences

The number and distribution of haplotypes were determined using the DnaSP software [34]. The haplotype network was built using the PopART software package [35] with the TCS network inference method, using the statistical parsimony method based on Templeton et al. [36] and Clement et al. [37] and implemented in TCS to visualize the mutation relationships. This approach identifies the number of haplotypes in the data and the number of evolutionary steps taken from one haplotype to another. TCS is useful in disentangling relationships at the intraspecific level. Finally, Tajima’s D was computed with the same software.

## 3. Results

From the 399 stored bovine epithelial tissue samples, 67 virus isolates (64 VSNJV and three VSIV strains) were successfully propagated in cell culture, amplified with RT-PCR, and sequenced for the P protein. Our sequences were uploaded to GenBank under the accession numbers ON567111-ON567177. See also Appendix A. Forward and reverse sequences were aligned using the UniPro Gene program and analyzed with the bioinformatic program MEGA-X together with the sequences of the P protein of 11 strains isolated from an outbreak in 2004, and described and deposited in GenBank by Sepúlveda et al. [16]. 

### 3.1. New-Jersey Phylogenetic Tree

The VSNJV phylogenetic tree based on the nucleotide sequences of phosphoprotein gene (P) built for our 2018 and 2004 isolates from Ecuador and sequences from the Americas (Figure 1) differentiated the strains into three supported clades (A, B, and C in Figure 1). The internal monophyletic group, with Indiana strains from Ecuador and Costa Rica as outgroups, showed Clade A with North American strains (USA and Mexico), then Clade B with Mexican and Central American strains (B1 and B2, respectively), and the internal Clade C with South American strains including the Colombian strains and also within clade C, two internal subclades composed of Ecuadorian isolates of 2004 that were isolated in six different provinces (Appendix A) and three internal Ecuadorian subclades (Ecuador-Andean/Coast, Ecuador-Andean and Ecuador-Amazonian) with our strains of 2018.

The 2018 epizootic subclade is formed by three lineages: the Amazonian and Andean with strong bootstrapping support (75 and 100%, respectively) and a small number of unsupported Andean and Coastal strains.

The Amazonian strains come from the provinces of Napo, Morona Santiago, Pastaza, Zamora Chinchipe, Orellana, and Sucumbios. The strains of the Andean clade were isolated in the provinces of Imbabura, Pichincha, Carchi, and Cotopaxi, and the coastal lowland strains were isolated in the provinces of Guayas, Los Ríos, Santo Domingo de Los Tsáchilas, and Esmeraldas.

### 3.2. Indiana Phylogenetic Tree 

Figure 2 shows the phylogeny of VSIV strains from Ecuador in comparison to additional sequences available from the Americas. Again, strains are region-specific in topology. The basal Clade A is comprised of northern and southern Brazil, the inner Clade B is formed by sequences from Argentina and southern Brazil, and Clade C by sequences from North, Central, and northern South America. Clade A corresponds to VSIV 2, Clade B to VSIV 3, and Clade C to VSIV 1. The strains from Ecuador are paraphyletic and have two different close ancestors to Clade C. The sequences from the Amazonian 2004 and 2018 (Loja and Orellana Provinces) strains have a Colombian ancestor, while the Andean-Coastal sequences (Pichincha and Santo Domingo de Los Tsáchilas) are located basally in Clade C and are closely related to the Brazilian and Argentinian Clade B. 

### 3.3. Haplotype Network Topology of VSV-NJ

The parsimony haplotype network (Figure 3) shows a structured haplotype geographic distribution. Low down in the network, the same phylogenetic tree sequence transformation series is observed from haplotypes of North America, and Central America, finishing with Colombian haplotypes to the left side. One of them (H4) is shared with the Ecuadorian sequence of the 2004 outbreak, with two additional unique sequences from Ecuador: H6 and H8 with three and six mutations, respectively. Then, high above in the network, the Colombian H3, which connects with the main Ecuadorian subnetwork as follows: 5, 7, and 10 mutations with Ecuador’s 2004 Amazonian outbreak (center in purple, from Orellana and Napo), Ecuador’s 2018 Amazonian outbreak (left in magenta) and 2018 Andean outbreak (right in yellow), respectively. Two unique Coastal haplotypes (H32 and H29 in orange) emerge from one mutation each from the most frequent haplotype in the Andes (H30). The 2018 outbreak haplotypes H27-Amazonian (left, magenta) and H36-Sierra (right, yellow) are closely and directly connected to the intermediate branch of the 2004 outbreak (H9, purple) in Ecuador using three and four mutations, respectively. 

## 4. Discussion

Vesicular stomatitis is known to be an endemic disease in North, Central, and South America, and outbreaks of the disease in Ecuador occur frequently. Between 2005 and 2014, Ecuador reported 182 outbreaks of VS to OIE-WAHIS (World Organization for Animal Health—Animal Health Information System). At that time, VS was a notifiable disease. However, due to the mild, self-limiting nature of the disease and improbable international spread through the trade of animals, VS was de-listed by the OIE as a reportable animal disease. Nonetheless, small VSV outbreaks have been reported every year in Ecuador, with a notable increase in 2018 with 399 registered outbreaks. Three factors likely favored the rapid spread of the disease in that year: the presence of vectors, climatic conditions, and, although self-quarantine measures were taken, the transport of cattle to auctions or between farms in the Andes highlands. In the next paragraphs, we discuss what are the likely causes (mechanisms) underlying transmission. 

### 4.1. VSNJV Epidemiology 

Concerning the VSNJV isolates, all 2018 strains from Ecuador are all closely related and can be grouped in one clade; clade C. In this clade, two subclades can be distinguished and these subclades are geographically separated: one with three lineages in the Amazon and the other that infects animals on the Pacific coast and in the Andean highlands. The presence of two subclades is possibly related to the fact that these regions are geologically separated by the Andes mountains that run from north to south, dividing the Ecuadorian mainland into three regions, from the west to the east: the Pacific coast, the Andean highlands, and the Amazon basin. (see Figure 4). Due to this geographic barrier, different virus populations may have diverged [38]. 

The analysis also showed several small and independent lineages in the Amazon region and the spread of another outbreak caused by a homogeneous genetic variant in the eastern coastlands and the Andes. It is well known that most stomatitis infections are caused by vectors [3,39,40] and the presence of several small, genetically variable strains in the Amazon region [41] indicates most possibly vector-borne transmission, perhaps through different insect species or populations or perhaps through wild mammals [39,42,43] but not due to contact between cows. Cow density is the lowest in this part of Ecuador, and most dairy farms are small-scale with only a few cows and little contact between cattle farms. We hypothesize that the outbreak in the eastern coastal lowlands and the Andes was most probably caused by insect bites, and was most likely transmitted by the movement of livestock. The outbreak in this area took place over a relatively short time of 3 months (September–November see Appendix A) and spread over several hundreds of kilometers. Cow density is high in this region and cattle trade is much more common. The highest density of cattle in Ecuador is found in the coastal and highland regions with 42.4% and 48.4% of livestock, respectively, while the remaining 9.2% is found in the Amazon [44]. There are no reports of VSV infection in other farm animals in the year 2018, making it highly impossible that other farm animals acted as vectors. However, we cannot exclude the possibility of VSV transmission via fomites or other passive vectors such as farm workers. If insects were involved in this outbreak, the genetic variation of the virus would have been more diverse due to the genetic adaptation of VSV to different insect reservoirs [45]. Moreover, insects are rare, especially in the Andes region with a much colder climate than the Amazon region, thus separating the Amazon region from the Andes. A low density of cattle in the Amazon means fewer contacts between animals, thus transmission is most likely via insect bites. In the coastal region, cattle trade and movement are more frequent, increasing the risk of the virus infection being transmitted between animals and facilitating a rapid spread of the virus over hundreds of kilometers. 

A comparison of the VSNJV sequences of Ecuador and other American countries shows that strains from Colombia and Ecuador have a high genetic homology, with more similarity in the provinces close to the border between the two countries. This indicates cross-border transmission of the virus between Ecuador and Colombia through cattle trade or the sharing of insect reservoirs. Additionally, the network topology suggests cross-border transmission or sharing of insect reservoirs (H4 shared, and H6, H8 closely connected), and an Ecuadorian 2004 variant, as a close ancestor of the 2018 outbreak. This is also shown in the intermediate branch of H9 and H7 Amazonian strains that connect with the 2018 Ecuadorian subnetworks See Figure 3. Of course, the haplotype network alone is not indicative of enzootic transmission in the Amazon. Many other factors such as geographical isolation and vector species play a role. The complete list of haplotypes and the countries where they were isolated is available in Appendix A. 

### 4.2. VSIV Epidemiology 

Concerning the epidemiology and transmission of VSIV strains, only limited information can be deducted from our analysis. We sequenced the P gene of only three strains from the 2018 outbreak. Only one other strain from Ecuador, from the outbreak in 2004, is available in GenBank. The phylogenetic tree topology of our Amazonian sequences is closely related to a Colombia sequence. The Andean sequences, basal in the clade, are derivates from Brazilian-Argentinian sequences. This suggests two different origins for the virus from Ecuador that emerged simultaneously in 2018, but also an unknown enzootic transmission cycle as a precursor of the epizootics.

The origin of the strains closely related to the Colombia sequence can be explained by the close economic relationship between the two countries (cattle trade) and sharing of insect reservoirs. However, the simultaneous occurrence of two lineages in the 2018 outbreak in the Andean mountains in the South-Central-North America clade requires further studies. 

### 4.3. Transmission by Vectors

Concerning the transmission of VSV by vectors, multiple studies have shown that different arthropods—such as *Lutzomyia* sand flies, *Simulium* black flies, and *Culicoides* biting midges—are competent vectors in the transmission of VSV-NJ. Suspect vectors with a very low transmission probability include *Aedes* mosquitoes [45,46,47]. The virus is also maintained in sandflies by transovarial transmission and in *Culicoides* by venereal transmission [6,48,49]. Moreover, experimental studies have shown that cows exposed to the bites of these arthropods are capable of generating antibodies against VSV [49,50,51]. 

In Ecuador, VSNJV was isolated in 1976 in the mosquito *Mansonia indubitans* [52,53], which breeds in lagoons with aquatic floating plants very common in the South American lowlands [54]. This supports the idea of several insect reservoirs present in the Ecuadorian Amazon causing regular, although not always detected, enzootic transmission with eventual emerging epizootics as in the 2004 and 2018 Amazonian outbreaks. Most probably several genetic variants of the virus exist in the Amazon. The outbreaks took place in counties separated by several hundred kilometers indicating that different gene pools of the virus could exist. Although the Influenza A virus is not transmitted by vectors, it has been shown for this virus that host and geographic barriers shaped the competition, extinction, and coexistence patterns of H1N1 lineages and clades [55]. Additionally, like the Venezuelan equine encephalitis virus, a different and enzootic vector can be present in the Amazon or the interface between forest and agriculture. It has also been demonstrated that the genetic diversity of the West Nile virus depended on the mosquito species [56,57]. If different hosts for the virus exist in the Amazon these hosts most probably will transmit different lineages. 

Therefore, systematic surveillance of potential sylvan vertebrates and insect vectors should be performed in those areas close to the epizootic areas to elucidate the mechanisms of emergence and re-emergence of VSV. The virus could also have spread thanks to the interaction with vertebrate animals, such as bats, deer, monkeys, and herbivorous rodents, which tend to serve as amplifiers since in these organisms, sustainable levels of viremia can occur [13,40,42,43]. 

## 5. Conclusions 

The two clades in the VSNJV phylogenetic tree of the 2018 outbreaks suggest a different transmission pattern in the Andean and Amazon regions. In the Amazon, the genetic diversity of the sequenced virus suggests transmission by vectors as a possible source of infection. In this region vector control measurements, to limit vector exposure and dispersion, are required to control viral spread [58]. Vector control is very important for the control of virus spread as has been shown for the management of the spread of VSV in equines in the USA [59].

In the Andean region, a group of genetically identical viruses was found, suggesting transmission of the virus through cattle movement. Because of the relatively cold climate in this region and the low density of insects, transmission from other reservoirs plays a minor role. Future clonal spread of the virus can be avoided when strict cattle movement restrictions are established at the first detection of infected animals. 

## 6. Recommendations

South American countries lack surveillance of VSV vectors and knowledge of reservoirs and vectors. We recommend carrying out vector and sylvatic vertebrate reservoir research and surveillance in tropical and subtropical areas where wild reservoirs can be a source of new outbreaks. 

In Ecuador, VS is a disease whose control is mainly based on self-quarantine. Additionally, a commercial bivalent vaccine (VECOL, Bogota, Colombia) is occasionally used in Ecuador to reduce viral spread. However, no efficacy studies are available for this vaccine. Therefore, studies should be planned to show the efficacy of this vaccine in overcoming the spread of the virus. Although there are limited data on the VECOL vaccine efficiency, other studies have shown protective immunity in cattle vaccinated on a commercial scale with inactivated, bivalent vesicular stomatitis vaccines [59]. It is also important to address how the vaccination status of neighboring countries might contribute to VSV mutations and spread in Ecuador. However, there is no public information about vaccination frequency in neighboring countries.

## 7. Limitations of the Study

Our analysis is based on 67 strains, only a small fraction of the 399 registered outbreaks in 2018. It is also possible that many more outbreaks took place in that year that were not reported. Thus, our sample is not a fair representation of the research population, possibly introducing a bias in our results.

To obtain enough viral material for this study, a cell culture of the VSV isolates, previously stored in a culture medium and stored in liquid nitrogen, was initiated. Cell culture propagation of the virus can introduce mutations that might not reflect the ancestral state of the original isolates. We examined the literature and it has been shown that, that only four single-nucleotide substitutions in 77,500 bases were found in the VSV genes after cell propagation (a mutation frequency of 5.16 × 10^−5^) [60], and our study is based on the sequencing of only about 27,470 bp. More indications that no mutations were introduced during cell propagation are the sequences of the strains circulating in the Andes. All 35 VSV strains, isolated on different time points and again grown in cell culture, amplified, and sequenced, had an identical sequence of the partially amplified P protein. We, therefore, believe that cell propagation did not introduce bias in our study. 

## Figures and Tables

**Figure 1 vetsci-10-00181-f001:**
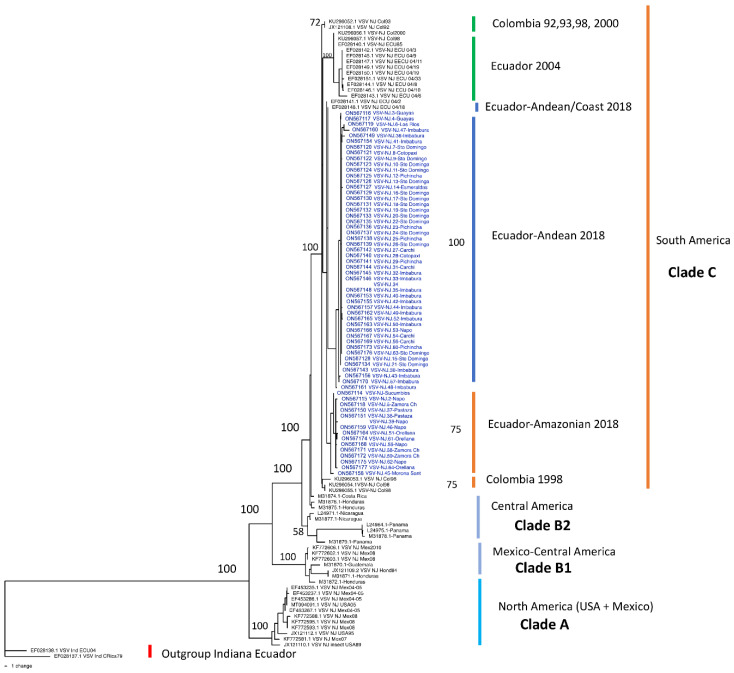
Phylogenetic tree based on the nucleotide sequences of phosphoprotein gene (P) of Vesicular stomatitis VSNJV isolates in Ecuador in 2018 and 2014. VSIV sequences from Ecuador and Costa Rica were used as an outgroup. The ML tree was inferred by method and Tamura 3-parameter model. The tree with the highest log likelihood (−3325.91) is shown. The robustness values for all the analyses were estimated using bootstrapping with 500 pseudo-replicates and are shown as percentages. Initial trees for the heuristic search were obtained automatically by applying the Maximum Parsimony method. Three strong, supported clades can be differentiated; Clade A with North American strains, Clade B with Mexican (B1) and Central American strains (B2), and Clade C with South American strains. Three subclades comprise the South American Clade, namely the basal Colombia strains and the Ecuadorian lineages: strains from 2004 and strains from the 2018 outbreak. The Ecuadorian epizootic strains show two supported lineages: Amazonian and Andean isolations and an unsupported Andean-Coastal strain.

**Figure 2 vetsci-10-00181-f002:**
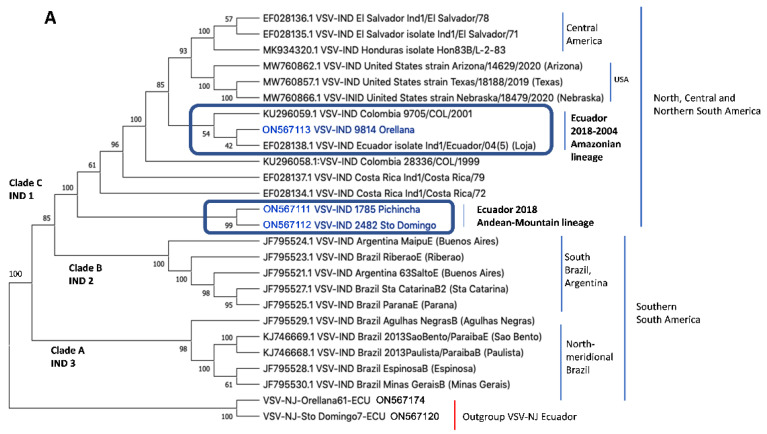
VSIV Phylogenetic Tree with GenBank strains and isolates from 2004 and 2018 from Ecuador. The tree was constructed using the maximum likelihood method with 500 bootstrap pseudo-replicates based on the Tamura-3 parameters model, gamma distribution (+G), and a number of discrete gamma categories of 5, and implemented by MEGA X. Clades A and B correspond to Brazilian and Argentina strains. Clade C includes three Ecuadorian isolates from 2018: Andean strains (1785 Pichincha and 2482 from Santo Domingo de Los Tsáchilas) and the Ecuadorian Amazon isolate (9814 from Orellana) together with a GenBank sequence from Loja in 2004 and a 2001 sequence from Colombia. This topology shows two different lineages in the Indiana VSV outbreak of 2018 in Ecuador.

**Figure 3 vetsci-10-00181-f003:**
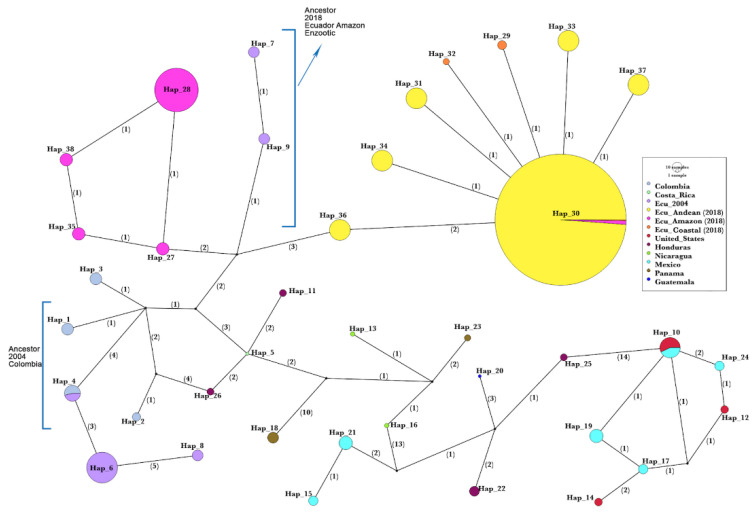
Haplotype network of VSV-NJ. Ecuadorian Isolates from 2004 are shown in a different color to those found in 2018. The number of mutations between each haplotype is shown (number in parenthesis). The haplotype size (circle) is relative to the frequency of each haplotype. Isolates from 2018 are color-coded according to the region of collection: yellow—Andes mountains; orange—Coast and magenta—Amazonian. The purple color corresponds to the 2004 Ecuador outbreak. According to the mutations between the group of isolates from 2004 and 2018, these have H7 and H9 as intermediate and closely connecting haplotypes, suggesting VSV enzootic interepizootic circulation variants. A common haplotype was found between the 2004 isolates and isolates from Colombia (H4), suggesting the Colombian origin of the 2004 epizootic. A complete list of the haplotypes and their origin can also be found in Appendix A.

**Figure 4 vetsci-10-00181-f004:**
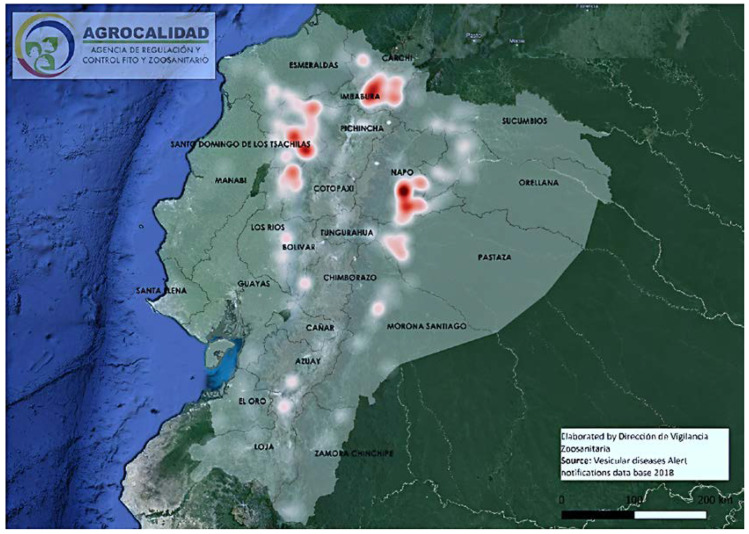
Map of areas with the highest number of VSV notifications in 2018. The intensity of the red color is relative to the registered cases of *VS*. The Andes mountains run from north to south and are indicated by a black dotted line. The Amazon provinces are located to the east and the coastal provinces are to the west of the Andes. The map is a modification of the map published by Salinas et al. [15].

## Data Availability

Our sequences were uploaded to GenBank under the accession numbers ON567111-ON567177.

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
