# Peer review of "Molecular Tracking of the Origin of Vesicular Stomatitis Outbreaks in 2004 and 2018, Ecuador"

_vetsci, 2023, doi:10.3390/vetsci10030181_

Round 1

Reviewer 1 Report (Previous Reviewer 2)

The manuscript entitled, "Molecular Tracking of the Origin of Vesicular Stomatitis Outbreaks in 2004 and 2018, Ecuador," describes the analysis of 67 VSV sequences from samples collected in 2018 in Ecuador. The sequences covered both VSV serotypes-VSNJV and VSIV- although VSNJV was largely overrepresented. The authors conclude, using phylogenetic construction and haplotype networks that there may be multiple sources that contributed to this VSV outbreak. Overall, the manuscript is well-written and the science is sound. I do have a few minor questions and issues with phrasing that need to be addressed (below). The topic of the manuscript is important in North and South America to increase our understanding of VSV transmission patterns. 

Comments:

1. Please rephrase the sentence in lines 78-80, it is wordy and difficult to understand. 

2. Lines 96-97: At what timepoints was the cell culture examined for CPE? And did you freeze down only if CPE was obvious?

3. Line 125: "of the P proteins" would be better phrased as "partial P protein sequence."

4. Since the phylograms were not discussed heavily and you include the phylogenetic trees in the main text, I would suggest removing supplementary figure 1 and 2. 

5. Did you know how many of the 399 samples were positive for VSV prior to virus isolation?

6. Figure 1: Can you identify the sequences generated in this manuscript better in figure 1?

7. Can you add genbank numbers to your sequences in figures 1 and 2?

8. Line 278: How many bootstraps were used for the tree displayed in Figure 2?

9. Lines 344-345: Weren't all of you sequences in a single clade (Clade C)? Please rephrase this sentence

10. Line 354-356: How did you come to the conclusion that there were several small outbreaks in the Amazon region from the VSNJV data? Please rephrase

11. Line 359: Please change 'different hosts' to vector-borne transmission

12. Line 368-370: There is not a set genetic difference that defines vector-borne versus non-vector-borne transmission. If the infection was fluctuating between insects and cattle multiple times, there would be a lot of genetic variation. However, I it is not accurate to say that insects were involved in one and not the other transmission patterns, especially with such limited number of sequences. Please rephrase

13. Line 373: "Transmission is mainly" should be changed to "transmission is most likely."

14. Line 389-391: You can't determine directionality from a tree, the virus could have originated (unnoticed) in Columbia entered Ecuador then remained dormant in Columbia until 2018. Please rephrase

15. Line 442: Were all of your samples in the study-defined Clade C? Do you mean within Clade C? Please rephrase

16. Line 481/ref 55: Were the mutations the reference identified in the P gene present in this studies sequences?

Author Response

Dear Reviewer, 

Thank you for your comments and assitance to improve this manuscript. Please see the attachment to know all the answers and changes. 

Kind regards, 

Reviewer 2 Report (Previous Reviewer 1)

The manuscript by Vasco-Julio et al. aimed to determine phylogenetic relationships among VSV isolates circulating in Ecuador during the 2004 and 2018 epizootics using sequences of the hypervariable phosphoprotein gene. The manuscript has significantly improved from earlier versions, and I recommend it for publication following minor revisions. 

1.     Add a citation supporting the statement in lines 52-54: “The ability of VSV to be transmitted through vectors is a great problem, since these vectors do not function only as simple means of transport, but also as a virus reservoir.”

2.     The sentence in the results (lines 171-174) is a description of the methods, not a result, so it should be moved accordingly. “The initial alignment matrix consisted of 614 bp and was trimmed to obtain a final alignment of 410 bp VSIV matrix. The VSVNJ sequences in the alignment matrix were trimmed to 422pb from an initial alignment of 642pb. The New Jersey and Indiana strains were separately analyzed.”

3.     In the discussion (lines 356-362), the authors reference cow-to-cow contact as a main to VSV spread in the Andes. I wonder what is the potential influence of VSV infections in other livestock animals, such as pigs and horses, or even the possibility of VSV transmission via fomites.

4.     The sentence influenza sentence at the end of the discussion (lines 428-429) might not be relevant to make a point about a Rhabdovirus transmitted by vectors. The discussion should be limited to taxonomically related viruses or viruses with similar patterns of transmission.

5.     Please fix the red-colored fund, mainly periods and parenthesis (i.e., lines 147, 166, 199, 434…) to make the text uniform. 

6.     Also, please fix the inconsistencies in spacing and figure references throughout the text (i.e., lines 178 &196; lines 351-352). 

Author Response

Dear reviewr, 

Thank you for your comments and assistance to improve this manuscript. Please see the attachment to know all the answers and changes. 

Kind regards 

This manuscript is a resubmission of an earlier submission. The following is a list of the peer review reports and author responses from that submission.

Round 1

Reviewer 1 Report

The manuscript by Vasco-Julio et al. aimed to determine phylogenetic relationships among VSV isolates circulating in Ecuador during the 2004 and 2018 epizootics. However, due to technical and methodological flaws, which are not addressed nor discussed, the authors are extrapolating the applicability of their results beyond what the design supports. I do not recommend this manuscript for publication for the following reasons.

First, the original field isolates used in this study were grown BHK-21 cells, and only cultures exhibiting CPE by 24 h were used for RNA isolation. It is well known that in the laboratory, VSV shows great capacity for genetic change and rapid adaptation, whereas in field conditions VSV remains relatively stable, with clear evolutionary patterns. Thus, cell culture propagation will undoubtedly introduce virus mutations that might not reflect the ancestral state of the original isolates. In addition, only using isolates exhibiting CPE in 24 h immediately removes those isolates that will fail to show signs of CPE in the 48-72 h period, introducing sampling bias. The need for cell-culture propagation is widely used to obtain complete or near-complete genome sequences for phylogenic analysis involving the five VSV proteins; however, the Vasco-Julio et al. performed an RT-PCR to amplify a partial segment of the phosphoprotein gene, which again may also introduce more sequence variability by the polymerase (which occurs even with low error rate enzymes). If an RT-PCR amplification approach was part of the study design, why not attempt to perform partial P amplification using RNA from the original isolates instead? 

For the subsequent analyses, previously published sequences were used to construct the phylogenetic trees. Some of those reference sequences were obtained from viruses first propagated in laboratory cell cultures (i.e., Sepulveda et al., Pauszek et al.) or directly obtained from the original field isolates (i.e., Velazquez-Salianas, Palinski et al., Fowler et al.). Moreover, not all the used sequences have the same length, with some being even shorter than the partial region amplified by Vasco-Julio et al. Still, there is no mention of how the aligned sequences were trimmed (or not), how gaps in shorter sequences were treated, and how cell culture propagation biases might (or not) influence the outcome of the analysis. Additional concerns regarding the construction of the phylogenic trees include using low bootstrapping, 500 pseudo-replicates instead of the standard 1000 as a measure of the analysis robustness, and the lack of branch length representing the evolutionary time between two nodes in both trees. 

Overall concerns in the discussion include the fact that the results do not support the conclusions, and most of the claims need to be more carefully addressed. As an example, the authors based their hypothesis of VSV emergence mechanism based “on vectors, reservoirs, ecological, and molecular genetic triggers” but only used very few partial sequences of viruses collected from infected livestock during two outbreaks. Moreover, the haplotype network alone is not indicative of enzootic transmission in the Amazon. Many other factors such as geographical isolation and vector species densities need to be addressed to construct a better elaborate (and supported) discussion. 

Lastly, in the recommendations, Vasco-Julio et al. suggest that a “future clonal spread of the virus can be avoided when strict cattle movement restrictions are established at the first detection of infected animals.” Not only does VSV spread not occur through “clonal spread,” but there is extensive literature about how viral spread during epizootics in U.S. and Canada persists despite quarantine and trade restrictions. VSV is also a vector-borne disease; vector control measurements to limit vector exposure and dispersion are also required to control viral spread. 

Additional general recommendations include:

·       Use valid nomenclature to name the respective VSV serotypes: either VSV-NJ and VSV-IN as extensively used in the literature or VSNJV and VSIV as indicated in the last International Committee on Taxonomy of Viruses (ICTV) update. 

·       Reported VSV biological vectors include: Lutzomyia sand flies (Diptera: Psychodidae), Simulium black flies (Diptera: Simuliidae), and Culicoides biting midges (Diptera: Ceratopogonidae). Suspect vectors with a very low transmission probability include Aedes mosquitoes (Diptera: Culicidae). Mosquitoes and Culicoides midges are very different taxonomic entities and should not be confused. 

·       The word “anthropoid” is used throughout different sections of the document. I believe the correct term the authors should refer to insect vectors is “arthropods” since “anthropoid” denotes primates (monkeys and apes). 

·       Mammalian reservoirs for VSV are yet to be elucidated. Autor needs to be more careful using this term and make a clear distinction between amplifying hosts, vectors that may act as overwintering reservoirs, and potential mammalian reservoirs. 

·       Although there is limiting data on VECOL vaccine efficiency, other studies are showing Protective immunity in cattle vaccinated on a commercial scale with inactivated, bivalent vesicular stomatitis vaccines. Also, it is important to address how the vaccination status of neighboring countries might contribute to VSV mutations and spread in Ecuador. 

·       Additional inaccuracies in the discussion include: VSV is NOT maintained in mosquitoes (Diptera: Culicidae), VSV transovarial transmission has ONLY been shown in Lutzomyia sand flies (Diptera: Psychodidae), venereal transmission has ONLY been shown in Culicoides biting midges (Diptera: Ceratopogonidae), and venereal transmission does not imply virus or transmission amplification. 

Reviewer 2 Report

The manuscript entitled, "Molecular tracking of the origin of Vesicular stomatitis outbreaks in 2004 and 2018, Ecuador," describes the phylogenetic analysis of the P-regions from 67 strains from the 2018 VSV outbreak in South America. Overall, the analyses were performed well, however, the quantity of experiments and analysis is low for a complete article. Additionally, the author makes claims that are not supported by the data presented, particularly in the discussion and conclusion sections. A list of concerns which is not complete is provided below.

Additional Major concerns: 

Line 181-183 How did you determine there were only 3 clades? There are many more supported clades in your phylogenetic tree.

Section 3.3: Please describe what the addition of the haplotype network tells us that the phylogenetic trees did not?

283-284: You cant make this claim with an n of 2 for number of sequences. 

Lines 286-288: You need more evidence than some putative clades on a gene tree to say that transmission occurred via a vector. Do you have other evidence to indicate that?

Line 290-292: Ref for higher genetic diversity of VSV in insects

Line 295-296: The density of cattle alone (and not other species that can transmit) does not confirm transmission via insect bites.

Conclusions: Many claims not supported by your data and/or references here. Need to completely rewrite.

Minor concerns:

Line 54: Please provide a reference for "the pest control decreases the probability..."

Overall methods: Too much detail. Please rephrase for clarity. (specifically lines 116-122 and phylogenetic analysis) 

Line 98: You examine for CPE right after infection? Please rephrase sentence for clarity

Line 128: Accessions should be in the data availability section and not here

Line 137: Please include the genbank accessions for VSV-NJ refs used for Indiana analysis

Line 263-266: What references do you have for this claim?

Lines 279-281: Ref for sentence?

Minor grammatical edits:

line 64 "live-stock" should be changed to livestock

line 80"remained" should be "remains"

line 81 "100-dreds" should be hundreds

Lines 89-91-please rephrase sentence for clarity

Line 91 start of sentence "the virus was grow" should be "the viruses were grown"

Line 95 "incubate" should be "incubated"

Line 159 "gen" should be gene

Line 258: "our country" should be changed to Ecuador